# Recent Progress on Genetically Modified Animal Models for Membrane Skeletal Proteins: The 4.1 and MPP Families

**DOI:** 10.3390/genes14101942

**Published:** 2023-10-15

**Authors:** Nobuo Terada, Yurika Saitoh, Masaki Saito, Tomoki Yamada, Akio Kamijo, Takahiro Yoshizawa, Takeharu Sakamoto

**Affiliations:** 1Health Science Division, Department of Medical Sciences, Shinshu University Graduate School of Medicine, Science and Technology, Matsumoto City, Nagano 390-8621, Japan; 2Center for Medical Education, Teikyo University of Science, Adachi-ku, Tokyo 120-0045, Japan; 3School of Pharma-Science, Teikyo University, Itabashi-ku, Tokyo 173-8605, Japan; saitou.masaki.nb@teikyo-u.ac.jp; 4Division of Basic & Clinical Medicine, Nagano College of Nursing, Komagane City, Nagano 399-4117, Japan; 5Division of Animal Research, Research Center for Advanced Science and Technology, Shinshu University, Matsumoto City, Nagano 390-8621, Japan; 6Department of Cancer Biology, Institute of Biomedical Science, Kansai Medical University, Hirakata City, Osaka 573-1010, Japan

**Keywords:** membrane skeleton, protein 4.1G, membrane palmitoylated protein, nervous system, bone formation, testis

## Abstract

The protein 4.1 and membrane palmitoylated protein (MPP) families were originally found as components in the erythrocyte membrane skeletal protein complex, which helps maintain the stability of erythrocyte membranes by linking intramembranous proteins and meshwork structures composed of actin and spectrin under the membranes. Recently, it has been recognized that cells and tissues ubiquitously use this membrane skeletal system. Various intramembranous proteins, including adhesion molecules, ion channels, and receptors, have been shown to interact with the 4.1 and MPP families, regulating cellular and tissue dynamics by binding to intracellular signal transduction proteins. In this review, we focus on our previous studies regarding genetically modified animal models, especially on 4.1G, MPP6, and MPP2, to describe their functional roles in the peripheral nervous system, the central nervous system, the testis, and bone formation. As the membrane skeletal proteins are located at sites that receive signals from outside the cell and transduce signals inside the cell, it is necessary to elucidate their molecular interrelationships, which may broaden the understanding of cell and tissue functions.

## 1. Protein 4.1 Family

### 1.1. Protein 4.1 in the Membrane Skeleton

Originally, membrane skeletal networks were found as a two-dimensional lattice structure beneath erythrocyte membranes, as schematically shown in Figure 1. Protein 4.1R–membrane palmitoylated protein 1 (MPP1)–glycophorin C is a basic molecular complex, in addition to ankyrin-band 3, attaching the actin–spectrin meshwork structures to form erythrocyte membrane skeletons, which support the erythrocyte membrane and provide stability, especially under blood flow [1]. Protein 4.1R (red cell) has 4.1–ezrin–radixin–moesin (FERM) and spectrin–actin binding (SAB) domains, and there are three other family members, namely 4.1B (brain), 4.1G (general), and 4.1N (nerve) [2,3]. In this review, we summarize recent studies on protein 4.1G in the peripheral nervous system (PNS) and bone development.

### 1.2. Protein 4.1G in PNS

Protein 4.1G was identified as FK506-binding protein 13 (FKBP13) [5]. We found its localization at two specific regions in Schwann cells that form myelin in the PNS: Schmidt–Lanterman incisures (SLIs) and paranodes [6]. Protein 4.1G assists in organizing internodes in the PNS [7], and is essential for the molecular targeting of MPP6 [8] and cell-adhesion molecule 4 (CADM4) [7] in SLIs. Thus, 4.1G–MPP6–CADM4, an analogous molecular complex to the erythrocyte membranes, exists in the PNS, likely functioning to resist external mechanical forces in SLIs [9]. 4.1G-deficient (-/-) mice showed motor impairment, especially with advancing age, and measurement of motor nerve velocity and the ultrastructure of myelin in the sciatic nerves demonstrated abnormalities under 4.1G-/- [10,11]. Considering that impairment of motor function with the tail-suspension test became worse after overwork treatment [11], careful attention is required in the rehabilitation of Charcot–Marie–Tooth (CMT) disease patients, which has been a controversial matter [12,13]. The SLI is thought to have function as a suspension structure against mechanical extension, similar to a spring [14], and in the case of 4.1G deficiency, the cell membrane may be destroyed. 

CADM4 is probably related to the myelin abnormality under 4.1G-/- because the localization of CADM4 in SLIs disappears in 4.1G-/- nerves [15]. Furthermore, CADM4-/- nerves exhibited similar structural changes to those observed in human CMT disease [15,16]. CADM4 depletion and subsequent disruption may be related to erbB2 because they interact with each other [17,18]. Recent reports have shown that CADM1 has a role in maintaining cell–cell interspaces to promote the proper function of gap junction proteins [19,20]. Other than CADM4, several proteins, such as AP3 complex, tubulin, heat shock cognate 71 kD protein, and 14-3-3 protein, have been found that relate to 4.1G, from immunoprecipitation studies in the retina [21,22]. Because various proteins are associated with 4.1 families [2,23], it is necessary to further elucidate the binding proteins and functions for 4.1G in the PNS.

Additionally, it remains unclear how actin–spectrin components are connected to the 4.1G–MPP6–CADM4 complex in the PNS, considering that actin abundantly forms filaments in SLIs [24]. Notably, the SAB domain is spliced in the retina [22], and another actin-binding peptide sequence was found in 4.1R near the common SAB domain in epithelial cells [25]. Thus, the relationship between 4.1G and the actin filaments in SLIs has not been clarified.

### 1.3. Protein 4.1G in Bone Formation

Bone structure is controlled by the balance between bone formation by osteoblasts and bone resorption by osteoclasts. Osteoblasts are differentiated from mesenchymal stem cells and preosteoblasts (osteoblast differentiation). Many factors, including hedgehog, parathyroid hormone (PTH), and Wnt, affect osteoblast differentiation [26]. Moreover, 4.1G regulates hedgehog-mediated bone formation and PTH receptor (PTHR) signaling [27,28,29,30].

The primary cilium is a hair-like immotile sensory organelle that possesses selectively distributed membrane receptors, such as G-protein-coupled receptors (GPCRs) and growth factor receptors, and ion channels on its surrounding membrane (ciliary membrane) [31]. The cilium is formed in various cell types during the G_0_ phase of the cell cycle. A hedgehog receptor (i.e., smoothened) is one of the typical ciliary GPCRs expressed in the stem/progenitor cells of various organs (e.g., blood vessels, bone, brain, breast, esophagus, gallbladder, heart, intestine, liver, lung, pancreas, and stomach) [32,33,34,35]. Smoothened participates in the proliferation and differentiation of the cells to control organogenesis and tissue homeostasis. 

Preosteoblasts form primary cilia on their surface. Deletion of the ciliary components, such as intraflagellar transport 80 (IFT80), IFT140, and kinesin 3a (Kif3a), disrupts preosteoblast ciliogenesis, ciliary hedgehog signaling, and femur or tibia formation [36,37,38]. Knockout of IFT20 in the cranial neural crest (CNC) disrupts ciliogenesis in CNC-derived osteogenic cells and leads to malformation of craniofacial bones [39]. These studies demonstrate the importance of primary cilia in bone formation. However, 4.1G is not recognized as a ciliary component, although it promotes ciliogenesis in preosteoblasts, as observed in the 4.1G-downregulated MC3T3-E1 preosteoblast cell line and 4.1G knockout preosteoblasts on trabecular bone in mouse new bone tibia [30]. In 4.1G-suppressed MC3T3-E1 cells, ciliary hedgehog signaling and subsequent osteoblast differentiation were attenuated, revealing a novel regulatory mechanism of bone formation by 4.1G.

Teriparatide, PTH-(1-34), is the first anabolic agent approved by the U.S. Food and Drug Administration for the treatment of osteoporosis [40]. Intermittent treatment with teriparatide facilitates osteoblast differentiation and suppresses osteoblast apoptosis [41,42]. Teriparatide activates PTHR, which is a GPCR. It strongly activates adenylyl cyclase (AC), produces cyclic AMP (cAMP) through G_s_ protein, and increases intracellular Ca^2+^ through G_q_ protein. In addition, 4.1G has been identified as an interacting protein of the carboxy (C)-terminus of PTHR [27]. Overexpression of 4.1G increases the amount of PTHR on the cell surface and PTHR-mediated intracellular Ca^2+^ elevation, suggesting that 4.1G augments the PTHR/G_q_ pathway by stabilizing the plasma membrane distribution of PTHR [27]. In contrast, PTHR/G_s_-mediated cAMP production decreases with 4.1G overexpression and increases with 4.1G downregulation [28,29]. Mechanistically, 4.1G binds to the N-terminus of AC type 6 and attenuates its activity [29]. These studies suggest that 4.1G alters the signal balance of PTHR, with a high 4.1G expression, G_q_ > G_s_, and with a low 4.1G expression, G_q_ < G_s_. It is necessary to investigate whether the regulation of the PTHR signaling balance by 4.1G is one of the mechanisms in the intermittent treatment of teriparatide. Moreover, the ciliary distribution of PTHR and its role in bone formation have been identified; PTH-related protein treatment and shear stress stimuli promote translocation of PTHR to primary cilia, and the ciliary PTHR mediates cell survival and osteogenic gene expression in osteoblastic and osteoclastic cells [43,44,45]. The role of 4.1G in ciliary PTHR signaling remains unclarified.

## 2. MPP Family

### 2.1. MPP in Membrane Skeleton

In erythrocytes, the 4.1R–MPP1 (a.k.a. p55)–glycophorin C (GPC) molecular complex stabilizes erythrocyte membranes [46]. MPP1 belongs to the membrane-associated guanylate kinase homolog (MAGUK) family, which is characterized by the presence of the postsynaptic density protein 95 (PSD95)/*Drosophila* disc large tumor suppressor (Dlg)/zonula occludens 1 (ZO1) [PDZ] domain, Src-homology 3 (SH3) domain, and catalytic inactive guanylate kinase-like (GUK) domain [47]. The PDZ and SH3 domains can interact with lipids and proteins. The SH3 domain also has intramolecular and intermolecular interactions with the GUK domain. The GUK domain is thought to have low enzymatic activity, although the binding site for ATP and GMP in MPPs is intact. Except for MPP1, there are two L27 (Lin2- and Lin7-) domains, in which MPPs are capable of interacting with each other. Additionally, MPPs have a HOOK/D5 domain that binds to protein 4.1 members, and there are seven family members [48]. MPP1 binds to two distinct sites within the FERM domain of the 4.1 family, and the alternatively spliced exon 5 in 4.1R is necessary for the membrane targeting of 4.1R in epithelial cells [49]. In addition to the protein–protein interaction, palmitoylation helps transport MPP family proteins to cell membranes, and enzymes known as zinc finger DHHC-domain-containing palmitoyl acyl transferase (zDHHC/PATs) have roles in palmitoylation [50]. In this review, we summarize recent studies on MPP6 and MPP2 in the PNS, CNS, and testis.

### 2.2. MPP6 in PNS

As mentioned previously, 4.1G-/- mice showed that protein 4.1G is essential for the molecular targeting of MPP6 and CADM4 in SLIs in the PNS, as shown in Figure 2a [7,8,9]. We evaluated what would happen if MPP6 itself was deleted [51]. MPP6 deficiency also resulted in the hypermyelination of peripheral nerve fibers, although the phenotypes, such as structural changes and impairment of motor function, were weak compared with 4.1G deficiency. 

The reason for hypermyelination without MPP6 was unclear. One of the MAGUK proteins, Dlg1 (SAP97), regulates membrane homeostasis in Schwann cells by interacting with kinesin 13B, Sec8, and myotubularin-related protein 2 (Mtmr2) for vesicle transport and membrane tethering [52]. The binding of the phosphatase and tensin homolog deleted on chromosome 10 (PTEN) to the specific PDZ domain of Dlg1 inhibits axonal stimulation of myelination [53], and this Dlg1–PTEN complex is thought to limit myelin thickness to prevent overmyelination in the PNS [54]. Conditional inactivation of Dlg1 in Schwann cells caused a transient increase in myelin thickness during development, suggesting that Dlg1 is a transient regulator of myelination [55]. Deletion of the Dlg1–PTEN complex increases Akt phosphorylation and subsequent hypermyelination in peripheral nerves [56,57,58,59]. Additionally, disruption of PTEN in Schwann cells results in hyperactivation of the endogenous phosphoinositide 3-kinase (PI3K) pathway, focal hypermyelination, myelin outfoldings, and tomacula [60]. Dlg1 interacts with Mtmr2 [61] in phosphatidylinositol (PI) lipid metabolism [62]. These signals probably regulate the interaction between the actin cytoskeleton and plasma membrane interplay in a phosphoinositide cascade [63]. In addition, increased phosphatidylinositol (3,4,5)-triphosphate (PIP3) causes membrane wrapping and myelination [64]. In MPP1-deficient neutrophils, PIP3 forms punctate aggregations, which result in abnormal pseudopods [65]. Thus, our findings suggest that MPP6-deficient nerves may be related to the PTEN/Akt signal pathway. 

The Src family of signal transduction proteins are also potentially related to the MPP family, because they interact with each other [66,67]. Additionally, as there are various PDZ-containing proteins in the PNS, such as MAGUK proteins (e.g., Dlg1 and MPP6), multi-PDZ domain protein 1 (MUPP1), pals-associated tight junction protein (PATJ), claudins, zonula occludens 1 (ZO1), and Par3 [68], but the extent to which they are interdependent or have mutual redundancy remains unclear.

### 2.3. MPPs and Lin7

#### 2.3.1. Lin7 in PNS (Figure 2a)

Mammalian Lin7 (a.k.a. Veli/Mals) that contains L27 and PDZ domains was originally identified in a protein complex with the potential to couple synaptic vesicle exocytosis to cell adhesion in rat brains, and there are three family members [69]. Localization of Lin7 was found in SLIs, and MPP6 mainly transported Lin7 to SLIs in the mouse PNS [51]. Interactions between the Lin7 and MAGUK families have been reported in various tissues, including MPP4 recruitment of PSD95 and Lin7c (Veli3) in mouse photoreceptor synapses [70], MPP7 formation in a tripartite complex with Lin7 and Dlg1 in MDCK culture cells, which regulates the stability and localization of Dlg1 to cell junctions [71], and MPP4 and MPP5 association with Lin7c at distinct intercellular junctions of the mouse neurosensory retina [72]. The L27 domain is a scaffold for the supramolecular assembly of proteins in the Lin7 and MAGUK families [73,74,75]. Originally, both Pals family proteins, MPP5 (Pals1) and MPP6 (Pals2), were identified as proteins associated with Lin7 [76]. Although MPP5 was also reported in the PNS [77,78], our finding indicates that Lin7 transport in the PNS is mostly dependent on MPP6.

#### 2.3.2. Lin7 in the CNS (Figure 2b)

In the cerebellum, high-resolution microscopic examination by Airy-confocal laser scanning microscopy revealed that the ring pattern in synaptic membrane staining and dot/spot areas inside synapses exhibited by Lin7 staining inversely correlated between MPP2+/+ and MPP2-/- synapses [79]. In MPP2-/- dendrites in cerebellar granular cells (GrCs), the Lin7-stained dot/spot areas did not overlap with the microtubule-associated protein 2 (MAP2)-stained dendritic shaft, indicating that MPP2 deficiency does not directly impair microtubule-based transport. In contrast, CADM1 exhibited a ring pattern in MPP2-/- synaptic membranes, and the number of Lin7-immunostained dot/spot areas localized inside the small CADM1-immunostained small rings was higher in MPP2-/- synapses than in MPP2+/+ ones. These results indicate MPP2 transports Lin7 from the dendritic shaft to postsynaptic membranes in synapses. Additionally, Lin7 was originally coimmunoprecipitated with CASK and Mint1, which bind to the vesicular trafficking protein Munc18-1 and are considered to play a role in the exocytosis of synaptic vesicles in presynaptic regions [69], whereas our findings demonstrated that Lin7 was abundantly localized at postsynaptic sites with MPP2 in GrCs in the cerebellum. 

#### 2.3.3. Lin7 in Testis (Figure 2c)

By immunohistochemistry (IHC), Lin7a and Lin7c were localized in germ cells, and Lin7c had especially strong staining in spermatogonia and early spermatocytes, characterized by staging of seminiferous tubules [80]. Lin7 staining became weaker in MPP6-/- testis according to both IHC and Western blotting, indicating a function of MPP6 in Lin7 transport in germ cells despite the unchanged histology of seminiferous tubules in MPP6-deficient mice compared with that of wild-type mice. In cultured spermatogonial stem cells maintained with glial-cell-line-derived neurotrophic factor, Lin7 was remarkably localized along cell membranes, especially at cell–cell junctions. Thus, Lin7 protein is localized in germ cells in relation to MPP6, which is a useful marker for spermatogenesis. 

#### 2.3.4. Proteins Interact with Lin7

Because MPP and protein 4.1 families are strongly related to Lin7 families, we listed the proteins associated with Lin7 from previous studies (Table 1) and categorized them into five groups. The first group is MAGUK family proteins and their relating proteins at cell–cell attaching sites, as described above in Section 2.3.1. The second group is the catenin–cadherin complex, an adhesion molecule. Aquaporin (AQP) 1 interacts with the Lin7–β-catenin complex in human melanoma and endothelial cell lines [81]. β-catenin and N-cadherin also interact with Lin7 in the rat brain [82], and the small GTPase Rho effector rhotekin interacts with the Lin7b–β-catenin complex in rat brain neurons [83]. In the third group, signal transduction proteins, such as the insulin receptor-substrate protein of 53 kD (IRSp53), are transported to tight junctions by Lin7 in cultured MDCK cells [84]. Signal transduction protein was detected at synapses in the rat cerebellum [85], and N-methyl-D-aspartate (NMDA) receptors increased in the IRSp53-knockout mouse hippocampus [86]. In the fourth group, synaptic proteins, such as GluN2B, bind to Lin7, and their complexes are carried by kinesin superfamily (KIF) 17 on microtubules in hippocampal neurons [87]. Interactions between the complex and PSD95 were also revealed in rat hippocampal postsynaptic regions [88]. 

In the fifth group, Lin7 interacts with several growth factor receptors. LET23 epidermal growth factor (EGF) receptor in *Caenorhabditis elegans* larval development [89] and Grindelwald tumor necrosis factor (TNF) receptor in *Drosophila* [90] are interesting examples, because they are related to the integration of cell signaling. Further examination of the Lin7 interaction with such receptors is necessary.

Concerning Lin7 knockout mice, although mice lacking Lin7a or Lin7c were viable and fertile, double knockout of mice for Lin7a and Lin7c was lethal before sexual maturation, suggesting that the functions of Lin7a and Lin7c likely compensated each other [91]. Additionally, Lin7a- and Lin7b-deficient mice are fertile and Lin7c was upregulated in mouse brain [92], indicating redundancy among Lin7 family members. Considering Lin7 in humans, disruption of cerebral cortex development by Lin7a depletion [93] and involvement in autism spectrum disorders by genetic alteration of *Lin7b* [94] has been reported. Therefore, target-cell-specific conditional disruption of Lin7 family proteins is required to elucidate the function of the Lin7 family.

**Table 1 genes-14-01942-t001:** Associated proteins to Lin7 families.

Protein Name	Category	Tissues and Cells	Method	Related Proteins	Functional Consideration	References
AQP1	2	Human melanoma WM115 and endothelial HMEC1 cell lines	IP, KD	β-catenin	AQP1-KD affects Lin7/β-catenin expression	[81]
BLT2 (Leukotriene B4 receptor)	5	MDCK cell line	PD, KD	CASK (Lin2) Mint (Lin10)	Transportation from the Golgi apparatus to the plasma membrane	[95]
BGT-1 (GABA transporter)	4	Recombinant Lin-7 and BGT-1 (PDZ target motif)	BC		Localization of transporter to plasma membranes	[82]
CASK (Lin2)	1	Recombinant CASK, Velis proteins, rat brain Mouse brain	IHC, YTH, IP	Mint (Lin10)	Synaptic plasma membranes, synaptic vesicle exocytosis to cell adhesion	[69]
CASK	1	Mouse brain	BC, PD	Mint (Lin10) KIF17	NR2B sorting vesicle carried by KIF17–Lin10 complex	[87]
Crumbs (*Drosophila*)	1	*Drosophila* eye under Lin7 mutation	IHC, PD	Stardust-PATJ	Light-dependent degeneration of photoreceptors	[96]
β-catenin	2	Recombinant β-catenin and Lin7a, MDCK cell line and rat brain lysate	BC, IP	E-cadherin	Cadherin–β-catenin adhesion complex	[82]
GluN2B (NMDA receptor)	4	Rat cerebral cortex, transfected NR2B or MALS	IP, PD	PSD95	MALS2 directly binds to NR2B	[88]
Grindelwald (*Drosophila*; TNF receptor)	5	Transfection of mutated Lin7	IHC	Stardust-PATJ-Crumbs	Transport of TNF (tumor necrosis factor) receptor	[90]
IRSp53	3	Rat brain, MDCK cell line	YTH, IP	SAP102	Formation or maintenance of the adhesion structure of epithelium	[97]
LET-23 (*C. elegans*; EGF receptor)	5	Transfection of mutated Lin-7	IHC, YTH	CASK (Lin2)	Vulval induction	[98]
LET-23	5	Transfection of mutated Lin-7	IHC	Lin2-Lin10 complex	Transport of LET-23 from the Golgi apparatus to the cell membrane	[89]
Mint (Lin10)	1	Rat homolog of the C. elegans Lin10	Cloning, IHC	CASK (Lin2)	Distributed in the membrane fraction in rat brain	[99]
MPP4	1	Porcine retinal membranes Transfection of bovine MPP4 L27C or L27N + C domain	IP, PD	MPP5	Veli3 and MPP4 most intense staining in photoreceptor terminals of the outer plexiform layer (OPL)	[72]
MPP5 (Pals1)	1	Cloning of Lin-7 binding partners	PD	MPP6 CASK (Lin2)	Localize to the lateral membrane	[76]
MPP6 (VAM1, Pals2)	1	Cloning of Lin7 binding partners	Cloning, PD	MPP5 CASK (Lin2)	Localize to the lateral membrane	[76]
MPP6	1	Transfection of human Veli1 binds to VAM1	PD		MPP6 does not bind to 4.1R	[100]
MPP7	1	Transfected human MPP7 L27C domain	PD	Dlg1	Enhanced localization of Dlg1 to cell junction	[71]
Rhotekin	2	COS7 cells and rat brain	YTH	PIST	Trafficking of protein in synapses	[101]
Stardust (*Drosophila*; Pals1)	1	Transfection of mutated Lin7	IHC	Crumbs	Transport of Grindelwalt (homologous to TNFR)	[90]

Category: Lin7-associating proteins are categorized into five groups as described in the text. BC: biochemical binding assay, IHC: immunohistochemistry, IP: immunoprecipitation, PD: pull down, YTH: yeast two-hybrid system.

### 2.4. MPPs and CADMs 

CADMs are Ca^2+^-independent adhesion molecules, and they have binding properties to both protein 4.1 and MPPs [102]. In the PNS, deficiency of the MPP6–Lin7 complex had little effect on CADM4, and cadherin and tight-junction proteins were retained [51]. However, scaffolding for CADM4 in SLI is mostly dependent on protein 4.1G, as shown in Figure 2a [15,16,51]. In testes, the expression and localization of CADM1 were retained in 4.1G/4.1B double-/- and MPP6-/- mice, as shown in Figure 2c [8,10,80]. 

In the CNS, scaffolding for CADMs is more complicated, because many MAGUKs are associated with CADM1 [103,104]. Although the PDZ domain of MPP2 was reported to directly interact with the C-terminus of CADM1 in rat hippocampal neurons [105], and nearly 80% of MPP2 dots overlapped with CADM1 areas by IHC and cerebellar lysate of MPP2 included CADM1 by immunoprecipitation study in our recent study in cerebellum, MPP2-/- synapses did not show reduction of CADM1 in cerebellar GrCs, as shown in Figure 2b [79]. Considering that CADM1-/- mice exhibited small cerebella with a decreased number of synapses compared with wild-type mice [106], the redundancy of MAGUK and 4.1 families to locate CADM family proteins has not been clarified.

### 2.5. MPP and Neurotransmitters

MPP2 specifically localizes to the cerebellar granular layer, particularly to dendritic terminals in GrCs facing the mossy fiber (MF) terminus at the cerebellar glomerulus, as schematically summarized with MPP2-interactive proteins in Figure 3a [79], because the MF–GrC synapses are the first place to transduce excitatory electrical signals into cerebellum [107]. MAGUK family proteins, such as PSD95 (Dlg4, SAP90), SAP102 (Dlg3), and Chapsyn-110 (Dlg2, PSD93), localize to both the molecular and granular layers [108]. To clarify the specific localization of MPP2, localizations of various MAGUKs are demonstrated in Figure 3b–k. Note that the gene loci of *MPP2* (in mouse chromosome 11) and *Dlg2* (in mouse chromosome 7) are different. 

MAGUKs are known to associate with excitatory NMDA and α-amino-3-hydroxy-5-methyl-4-isoxazole-propionic acid (AMPA) glutamate neurotransmitters [109,110]. At MF–GrC synapses, NMDA receptors such as GluN1 and GluN2A/C were detected [111] as well as adherens junctions consisting of GrC dendrites [112]. Additionally, transmembrane AMPA receptor regulatory proteins (TARPs) γ2/γ7 were detected in the postsynaptic regions of MF–GrC synapses with AMPA receptors GluA2/GluA4 [113]. GrC-specific GluA4-knockout mice showed a delay in eyeblink conditioning, but not locomotor coordination [114]. Motor dysfunction with a simple walking test has not been detected in MPP2-/- mice [79], and further examination for the neurological examination under conditioning is necessary.

In addition to excitatory neurotransmitters, a recent study demonstrated that MPP2 interacts with inhibitory γ-amino butyric acid (GABA) neurotransmitters without the involvement of gephyrin in rat hippocampal neurons [115]. In the cerebellar glomerulus, GABAergic neurotransmission is mediated between Golgi cells and GrCs, and two types of GABAergic inhibition have been proposed: phasic and tonic inhibition [116]. For phasic inhibition (transient inhibition), GABA_A_R consists of α1, α6, β2/3, and γ2 subunits in synaptic regions, and, for tonic inhibition (sustained inhibition), GABA_A_R consists of α1, α6, β2/3, and δ2 subunits in extrasynaptic regions [116]. GABA_A_Rs in synaptic regions interact with neuroligin 2, GABA_A_R regulatory Lhfpl, gephyrin [117], and synaptic scaffolding molecule (S-SCAM)/membrane associated guanylate kinase 2 (MAGI2) [118].

Figure 4 shows double-immunostaining and Airyscan-confocal laser scanning microscopy observations, demonstrating comparative localizations of MPP2 to α1, gephyrin, and α6. Gephyrin is a scaffold protein in the synaptic region, and α6 is a GABA_A_R in the extrasynaptic region. α1 (Figure 4a,c,f,h,k,m) and α6 (Figure 4l,m) staining were observed as dot/line patterns, whereas MPP2 (Figure 4b,c) and gephyrin (Figure 4g,h) staining were recognized as dot patterns in CG. Approximately 44% of the α1-stained areas (*n* = 94) overlapped with the MPP2-stained dots (Figure 4e), indicating a relationship between α1 and MPP2. Additionally, ~43% of the gephyrin-stained dots (*n* = 65) overlapped with α1-stained areas (Figure 4j), and ~27% of the α1-stained areas (*n* = 74) overlapped with α6-stained areas (Figure 4o). Thus, the overlap of the α1/MPP2 areas with the gephyrin and α6 areas indicate that α1/MPP2 localize in synaptic and extrasynaptic regions, respectively.

As MPP2 was reported to interact with several GABA_A_R subunits [115] and various subunits are present in the cerebellum [119], it is necessary to consider the interdependence of the GABA_A_R subunits. In the thalamus of the α4-knockout mouse, δ was decreased, whereas α1 and γ2 were increased in extrasynaptic regions, suggesting compensation among GABA_A_R subunits [120]. In addition, in the α1-knockout mouse, increases in the α3, α4, and α6 subunits, reductions in the β2/3 and γ2 subunits, and maintenance of the α5 and δ subunits were reported [121]. Further studies on the balance of these GABA_A_R subunits under MPP deficiency are necessary.

Several membrane skeletal proteins have been reported to interact with GABA_A_R. A giant ankyrin-G controls endocytosis of GABA_A_R by interacting with GABA_A_R-associated protein (GABARAP) in the mouse-cultured hippocampus [122]. GABA_A_Rα5 interacts with a membrane skeletal ezrin–radixin–moesin family protein, radixin, in mouse hippocampus [123]. GABA_A_R also interacts with neuroligin1 and CASK in inhibitory neuromuscular junctions in *C. elegans* [124]. MPP2 may be dependent on these membrane skeletal proteins to locate GABA_A_R.

### 2.6. MPP Families in Synapses

MAGUK proteins become oligomers because of PDZ–SH3–GUK tandem domains, function as a molecular complex in cell membranes specifically at cell–cell adhesion areas, and occur in various tissues and organs [125,126]. Particularly, there are many MAGUK family proteins in synapses, which function in postsynaptic density formation and signal transduction, and their impairment is related to some mental diseases [110,127,128,129,130]. A recent genome-wide association study (GWAS) also demonstrated the relationship between MPP6 and various psychiatric disorders: the *MPP6* gene was included in 64 genome loci for bipolar disorders compared among European ancestry [131], in 109 genome loci associated with at least two psychiatric disorders including anorexia nervosa, attention-deficit/hyperactivity disorder, major depression, obsessive–compulsive disorder, schizophrenia, and Tourette syndrome [132], and in 108 genome loci for schizophrenia patients [133]. *MPP6* was also included in 57 hard sweep genes after the initial movement of the evolutionarily recent dispersal of anatomically modern humans out of Africa, among genes related to biological processes, including ciliopathies, metabolic syndrome, and neurodegenerative disorders. [134]. In addition, a GWAS for sleep disorders demonstrated novel genome-wide loci on human chromosome 7 between *NPY* and *MPP6*, and disruption of an ortholog of MPP6 in *Drosophila melanogaster* was identified in sleep center neurons relating to decreased sleep duration [135]. In these respects, it is necessary to evaluate neurological psychological impairments in genetically modified MPP-deficient mice, which may be related to human diseases that are caused by mutation in *MPP* genes.

## 3. Conclusions

The 4.1 and MPP families are not only membrane skeletal components but are also widely distributed in various organs to transport intramembranous and signal transduction proteins. Especially, 4.1G has an obvious function in myelin formation in the PNS. There may be some interdependence and redundancy among the 4.1 and MPP families, as well as related proteins in other organs such as the CNS and testis, which brings about future challenges to examining cross-breeds of several genetically modified model mice. Considering that the molecular evolution of vertebrate behaviors may be related to the diversity of MAGUK proteins including MPPs [136], further evaluation of a wide range of molecular complexes, by proteomic and transcriptome analyses combined with genetically modified animal models, may broaden the understanding of normal morphological and physiological functions as well as physical and mental impairment. 

## Figures and Tables

**Figure 1 genes-14-01942-f001:**
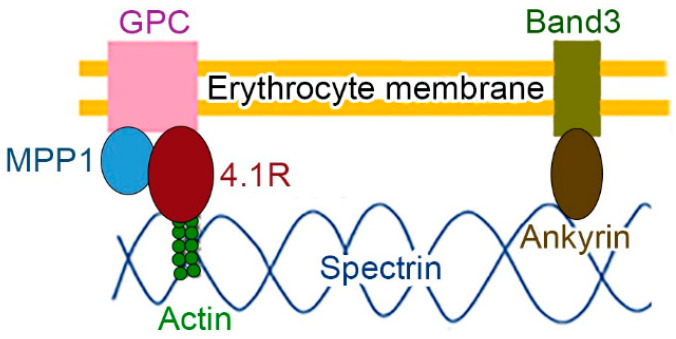
Schematic representation of an erythrocyte membrane skeleton. The spectrin–actin network structure is connected by protein 4.1R-membrane palmitoylated protein 1 (MPP1) and ankyrin to the intramembranous proteins glycophorin C (GPC) and band 3, respectively. The concept was obtained from previous research [4].

**Figure 2 genes-14-01942-f002:**
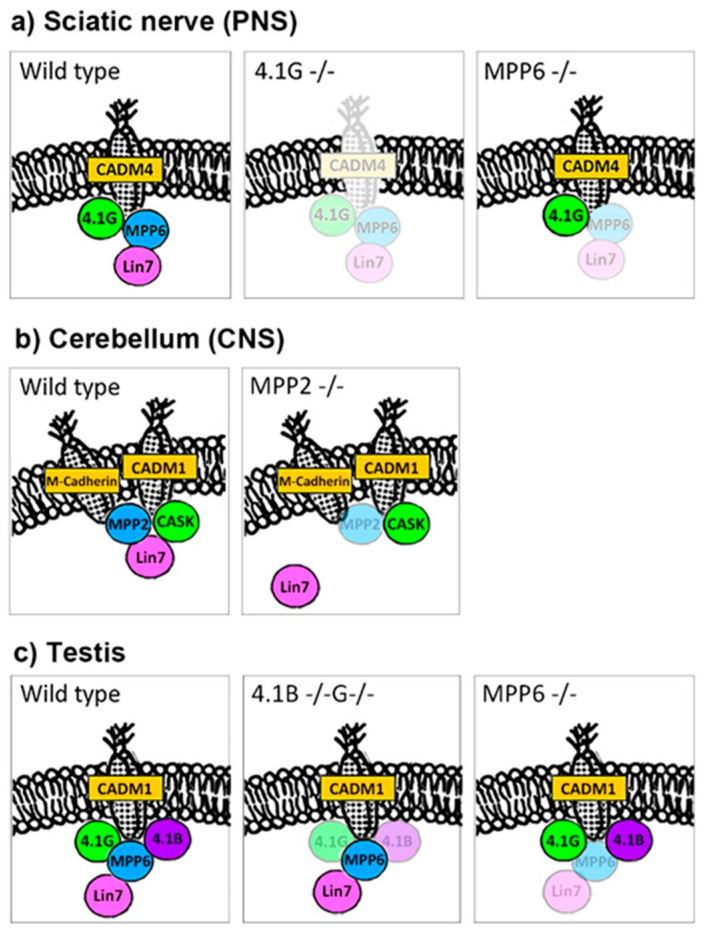
Schematic representation of the relationships among membrane skeletal proteins (4.1, MPP, and CADM) in the PNS (**a**), CNS (**b**), and testis (**c**). Note the different interdependences among those proteins in different organs, revealed by the genetic depletion of the proteins. The picture is partially modified from a previous paper [51].

**Figure 3 genes-14-01942-f003:**
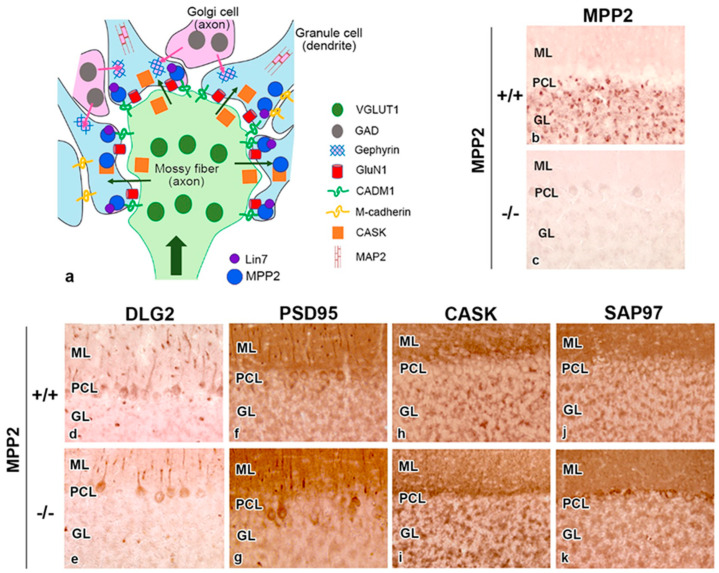
(**a**): Schematic representation of MPP2-relating proteins in the cerebellar glomerulus. Note that MPP2 interacts with various adhesion molecules, such as CADM1 and M-cadherin, as well as signal transduction proteins such as CASK and Lin7. GAD: glutamic acid decarboxylase, VGLUT1: vesicular glutamate transporter 1. The picture is partially modified from a previous paper [79]. (**b**–**k**): Localization of MAGUKs (MPP2 (**a**,**f**), DLG2 (**b**,**g**), PSD95 (**c**,**h**), CASK (**d**,**i**), and SAP97 (Dlg1) (**e**,**j**)) in the cerebellar cortex in MPP2+/+ (**a**–**e**) and MPP-/- (**f**–**j**) mice. Note that MPP2 is mainly observed in the granular layer (GL). ML: molecular layer, PCL: Purkinje cell layer.

**Figure 4 genes-14-01942-f004:**
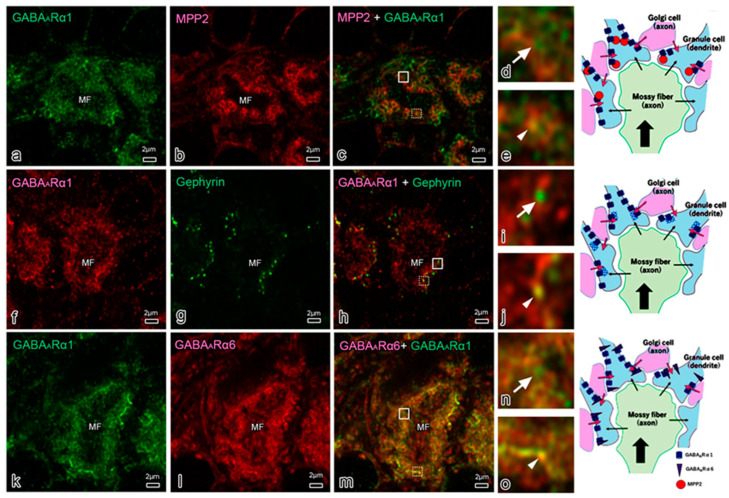
Comparative localization of GABAARα1 (**a**,**f**,**k**) with MPP2 (**b**,**c**), gephyrin (**g**,**h**) and GABAARα6 (**l**,**m**) in mouse cerebellar glomeruli. Examples of two-color overlapping regions are shown in (**d**,**e**,**i**,**j**,**n**,**o**) from areas in pictures (**c**,**h**,**m**), respectively. Detailed count data regarding the overlap is described in the text. The right lane demonstrates a summarized schematic drawing of their localizations obtained by immunohistochemistry; it does not consider how to make GABA_A_R with five subunits.

## Data Availability

Not applicable.

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
