# Peer review of "Recent Progress on Genetically Modified Animal Models for Membrane Skeletal Proteins: The 4.1 and MPP Families"

_genes, 2023, doi:10.3390/genes14101942_

Round 1

Reviewer 1 Report

In table 1, since the proteins associated with Lin7 are categorized into five groups, it would be helpful to group the Lin7-associated proteins in Table 1 and to clarify the group numbers. This would make the table more organized and easier to read.

I would suggest adding a discussion or summary of the functional roles of 4.1 and MPP families in the peripheral nervous system, central nervous system, testis, and bone formation to section 3. This would provide the reader with a better understanding of the importance of these proteins and the potential applications of the genetically modified animal models.

The authors may want to consider adding a section on the challenges and limitations of using genetically modified animal models to study 4.1 and MPP family proteins. This would be a valuable addition to the article, as it would help potential users to assess the strengths and weaknesses of these models.

The authors may also want to consider adding a section on the potential therapeutic applications of the genetically modified animal models. This would be a good way to highlight the potential benefits of this research for patients with diseases that are caused by mutations in 4.1 and MPP family genes.

There are some run-on sentences, for example: the run-on sentence from lines 186 to 188 can be fixed by splitting it into two sentences. 

Author Response

Thank you very much for your constructive comments. I revised the manuscript according to your suggestions. The revised parts are highlighted as red in the revised text.

  1. Table 1: As you suggested, a line for the category was added and clarify the group numbers.
  2. Lines 374-375 and 378-382:  As you suggested, summary of the functional roles of 4.1 and MPP families were added in Concluding remarks. Additionally, description about consideration about the challenges of using genetically modified model mice to sutdy 4.1 and MPP family proteins was added.
  3. Line 186: As you suggested, run-on sentence was improved.

Reviewer 2 Report

The authors investigated the molecular interrelationship between Protein 4.1 and membrane palmitoylated protein (MPP) families in different organs, and found that MMP2 is specifically localized to the cerebellar granular layer, where it is associated with GABAAR. The manuscript is well written and interesting. However, there are some minor points as follows.

In Table 1, it is better to align the lines on the top.

The quality of Figure 3b-k is not good. The authors had better clarify the focus in Figure 3.

Author Response

Thank you very much for your constructive comments. I revised our manuscript according to your suggestion. The revised parts are highlighted as red in the revised text.

  1. Table1: As you suggested, lines on the top were aligned.
  2. Figure 3: As you suggested, resolution of pictures was improved.